# Sustainable Antibacterial and Antiviral High-Performance Copper-Coated Filter Produced via Ion Beam Treatment

**DOI:** 10.3390/polym14051007

**Published:** 2022-03-02

**Authors:** Sunghoon Jung, Jun-Young Yang, Donghwan Jang, Taeyoon Kim, Ki Ho Baek, Hyunkyung Yoon, Joo Young Park, Sang Kwon Kim, Jinhyuk Hong, Sungweon Ryoo, Ho Won Jang, Seunghun Lee

**Affiliations:** 1Department of Nano-Bio Convergence, Korea Institute of Materials Science, Changwon 51508, Korea; hypess@kims.re.kr (S.J.); yjy8184@kims.re.kr (J.-Y.Y.); kihoback@kims.re.kr (K.H.B.); yoyoy823@kims.re.kr (H.Y.); jypark@kims.re.kr (J.Y.P.); 2Department of Materials Science and Engineering, Seoul National University, Seoul 08826, Korea; 3Department of Organic Material Science and Engineering, Pusan National University, Busan 46241, Korea; 4Clinical Research Center, Masan National Tuberculosis Hospital, Changwon 51755, Korea; dhjang0120@korea.kr (D.J.); rlaxodbs92@korea.kr (T.K.); viweon@korea.kr (S.R.); 5Airo Co., Ltd., Goyang 10251, Korea; airokorea@hanmail.net; 6Atomy Co., Ltd., Gongju 32543, Korea; jhhong@atomy.kr; 7Department of Materials Science and Engineering, Research Institute of Advanced Materials, Seoul National University, Seoul 08826, Korea; hwjang@snu.ac.kr

**Keywords:** ion beam, fiber, filter, antibacterial, antiviral, copper

## Abstract

With the spread of severe acute respiratory syndrome coronavirus 2 (SARS-CoV-2), disease prevention has become incredibly important. Consequently, mask and air-purifier use has increased. The filter is the core component of these items. However, most filter materials lack antimicrobial properties. Copper is a sustainable antimicrobial material. When copper is deposited onto the filter’s surface, the microorganisms that come into contact with it can be effectively inactivated. In this study, we used an oxygen ion beam with a controlled process temperature to treat filter surfaces with copper. This enabled a strong adhesion of at least 4 N/cm between the copper and the filter fibers without damaging them. Upon exposing the filter to bacteria (*Staphylococcus aureus* ATCC 6538, *Klebsiella pneumoniae* ATCC 4352, *Escherichia coli* ATCC 25922, and *Pseudomonas aeruginosa* ATCC 27853) for one hour, a >99.99% removal rate was attained; when the filter was exposed to SARS-CoV-2 virus for one hour, it inactivated more than 99%. These beneficial properties minimize the risk of secondary infections, which are significantly more likely to occur when a conventional filter is replaced or removed.

## 1. Introduction

Recently, severe acute respiratory syndrome coronavirus 2 (SARS-CoV-2), first discovered in Wuhan, Hubei Province, China, has spread rapidly worldwide, raising widespread health concerns. The World Health Organization (WHO) declared the coronavirus disease of 2019 (COVID-19) a pandemic on 12 March 2020. Over two years after the outbreak started in December 2019, over 300 million people have been infected, over 5 million people have died worldwide, and the infections and mortality rates have continued to increase [1]. Materials with antiviral properties have been used to prevent the spread of infectious diseases caused by these viruses. For example, ethanol-based disinfectants are the most commonly used sanitizers to treat humans, while chlorine-based and ammonium-based disinfectants are used for disinfecting living spaces [2,3]. These disinfectants effectively remove SARS-CoV-2 and other microorganisms but should be reapplied after use.

In contrast, copper is a sustainable antimicrobial material proven to inactivate viruses even on the nanometer scale [4,5,6,7,8,9,10,11]. Recent studies have reported on the antiviral performance of copper against SARS-CoV-2. For example, Hutasoit reported that 96% of SARS-CoV-2 viruses placed on copper-coated steel plates were inactivated within two hours; however, no viruses were inactivated after five hours of contact with uncoated stainless steel [12]. A study on SARS-CoV-2 inactivation as a function of the type of surface the virus contacts was conducted by Doremalen in 2020. They measured the time dependence of the titer of SARS-CoV-2 after it was placed on cardboard, stainless steel, plastic, and copper surfaces. On the copper surface, no viable SARS-CoV-2 was observed after four hours, whereas SARS-CoV-2 was observed on the surface of the other materials even after twenty-four hours [13].

A key material used in masks and air purifier filters is the polymer membrane, which is an assembly of one-dimensional fibers. Air can pass through small pores in the membrane, but the fibers block droplets, bacteria, and dust. Nevertheless, while the polymer membrane can filter bacteria and viruses, the latter may remain on the membrane surface. This can cause secondary infections when the filters are replaced. Guo reported in 2020 that bacteria caught in high-efficiency particulate absorbing (HEPA) filters can survive longer than in dust. It was noted that bacteria in HEPA filters fill an ecological niche that may have been neglected in indoor environments [14].

A solution to this problem is the application of a copper coating to antibacterial and antiviral filters. However, it is essential to adhere the copper to the surface of the filter firmly because if the copper detaches from the filter, a human can inhale the copper, which becomes a toxin when inside the body [6]. In previous works, modifications to the surface of a filter using an ion beam enabled the strong adherence of materials onto the filter [15,16,17,18]. The adhesion was accomplished without damaging the fibers of the filter. We found that SARS-CoV-2 was inactivated on the copper-coated mask within one hour [19].

In this study, we observed that the ion beam surface treatment was very effective in improving the adhesion between the copper and the filter fibers. We found that the copper-coated filter inactivated more than 99.99% of four examples of bacteria (*Staphylococcus aureus* ATCC 6538, *Klebsiella pneumoniae* ATCC 4352, *Escherichia coli* ATCC 25922, and *Pseudomonas aeruginosa* ATCC 27853) and inactivated more than 99.8% of SARS-CoV-2.

The remainder of this paper is organized as follows. Section 2 explains the materials and methods we employed. Section 3 discusses the main results, and Section 4 describes the conclusions.

## 2. Materials and Methods

### 2.1. Materials

A polyethylene terephthalate (PET) (Airo Co., Ltd., Goyang, Korea) filter with an average diameter of 30 μm and a surface density of 70 g/m^2^ was used.

### 2.2. Ion Beam Treatment and Copper Sputtering Deposition

Figure 1a shows a schematic of the ion beam treatment and copper sputtering processes. The process chamber is lab-made and includes the ion beam source and direct-current magnetron sputtering source. After the chamber vacuum reached 1 × 10^−5^ torr of pressure, we proceeded with the subsequent process. First, oxygen gas was injected. Oxygen ion beams were generated from a linear ion beam source. The process was performed by varying the applied voltage, gas flow rate, and sample stage speed such that the total applied energy density was 1.45, 3.13, and 8.84 J/cm^2^. The latter three conditions were named Ion Beam 1, Ion Beam 2, and Ion Beam 3, respectively. Next, we measured the maximum temperature of the process by attaching labeled temperature-measuring tape (3E-50, 3E-70, 3E-90, 3E-110; Nichiyu Giken Kogyo Co., Ltd., Tokyo, Japan) to the sample stage. After the ion treatments were performed, copper was deposited using a direct current magnetron sputtering system with a purity target of more than 99.99%. Argon gas (purity > 99.99%) was injected with 30 sccm to apply a working pressure of 1.0 mTorr, and the copper was deposited with a power density of 2.55 W/cm^2^ at a speed of 0.6 m/min in four separate runs. The thickness of the copper film was 30 nm. Detailed ion beam and sputtering process conditions are in Appendix A.

### 2.3. SRIM Calculations

Stopping and Range of Ions in Matter (SRIM) is software (SRIM; Version 2013, Liverpool, MD, USA) that, as the name suggests, calculates the stopping and range of ions in matter [20,21,22,23]. This code can be used to calculate the amount of damage caused by ion irradiation. We used SRIM to calculate the number of phonons generated when oxygen ions collide with PET. In this study, there were two limitations in the SRIM calculations: (1) Oxygen is a diatomic molecule and can be decomposed into atoms when colliding with the PET surface, and (2) when oxygen ions collide with PET, a chemical reaction may be induced because oxygen is a reactive gas. Therefore, we made the following two assumptions to simplify the calculation of the relative number of phonons generated in the collisions: (1) when molecular oxygen ions collide with PET, the molecules decompose into two monatomic oxygen ions with half the energy [24], and (2) there are no chemical reactions between oxygen ions and PET substrates.

The density of the PET was 1.397 g/cm^3^, and the emitted ion energies were 180, 300, and 600 eV for Ion Beam 1, Ion Beam 2, and Ion Beam 3, respectively [17,18]. The relative number of phonons generated was calculated based on the ion energy of Ion Beam 3, with ion fluences of 2.52 × 10^16^, 3.26 × 10^16^, and 4.60 × 10^16^ ions/cm^2^ for Ion Beam 1, Ion Beam 2, and Ion Beam 3, respectively.

### 2.4. Adhesion Test of Copper Deposited onto the Filter

The test involving the T-peeling of tape (3M VHB^TM^ 4910 Tape; 3M, St. Paul, MN, USA), illustrated in Figure 1b, was used to evaluate the adhesion force between the sputtered copper film and the filter fibers. After attaching tape to the copper deposited onto the filter, which was 10 mm wide, the end of the filter was fixed, and the edge of the tape was pulled directly away from the filter at a 90° angle (forming a T-shape) to evaluate the peeling.

### 2.5. Observing the Surfaces of the Filters and Tapes

Filter samples with a size of 100 mm × 100 mm were prepared to measure the rate of change of the area of the filters. After completing the ion beam treatment described in Section 2.2, the filter samples sizes were measured. We took digital pictures of the filter samples before and after the surface treatment on the grid, then calculated the change in their areas using commercial graphic drawing software (Rhinoceros 3D; Version 6.0. Robert McNeel & Associates, Seattle, WA, USA).

An optical microscope (ECLIPSE LV150N; Nikon, Tokyo, Japan) and field-emission scanning electron microscopy (FE-SEM; JSM 6700F, JEOL, Tokyo, Japan) were used to observe the surfaces of the filter and the detached tape.

### 2.6. Method for Evaluation of Antibacterial Performance

The copper-coated filter’s antibacterial properties were evaluated, according to the KS K 0693:2016 test method, using the following bacteria: *Staphylococcus aureus* ATCC 6538, *Klebsiella pneumoniae* ATCC 4352, *Escherichia coli* ATCC 25922, and *Pseudomonas aeruginosa* ATCC 27853. The reduction rate was calculated using
(1)reduction rate (%)=(1−B/A)×100
where *A* is the colony-forming unit (CFU) per mL of the control group, and *B* is the experimental group.

### 2.7. Method for Evaluation of SARS-CoV-2 Elimination Performance

Figure 1c shows a schematic of the test system used to evaluate the SARS-CoV-2 elimination performance. For the aerosol test, a closed cylindrical chamber was produced using a vibrating nebulizer (HL100A; Health & Life Co., Ltd., New Taipei City, Taiwan). A 30 nm-thick copper-coated filter with a 70 mm diameter was installed in the chamber, and bioaerosols with SARS-CoV-2 (NCCP43326, National Culture Collection for Pathogens, Cheongju, Korea) in 2% fetal bovine serum containing Dulbecco’s modified Eagle’s medium (DMEM) (2.87 × 10^6^ plaque-forming units (PFU)/mL) were sprayed onto the filter at a flow rate of 320 μL/min for 300 s. For comparison, the same process was repeated using a filter that was not coated with copper. The filters were then immersed in 10 mL of DMEM for two minutes to separate the virus particles. The plaque assay was conducted in Vero76 cells (CRL-1587; American Type Culture Collection, Manassas, VA, USA) following the protocol described in a previous study [25]. The experiments with live SARS-CoV-2 were conducted at the biosafety level three laboratory in the Masan National Tuberculosis Hospital.

## 3. Results and Discussion

### 3.1. Condition of the Filters after Ion Beam Treatment

The surfaces of the filters were checked for changes to the filter fibers caused by the oxygen ion beam irradiation. Figure 2a shows the rate of change of the area of the 100 mm × 100 mm filter sample after treatment with the ion beams. Ion Beams 1 and 2 caused area reduction rates of 0.33% and 0.46%, respectively. The area reduction rate caused by Ion Beam 3 was 3.81% (more than eight times that of Ion Beams 1 and 2), and the filter also contracted by approximately 2% in the longitudinal direction. The maximum process temperatures for Ion Beams 1 and 2 were 50 °C and 75 °C, respectively. In contrast, the process temperature for Ion Beam 3 was approximately 120 °C. The number of phonons generated was calculated using SRIM, and the relative amounts of generated phonons by ion beam treatment are shown in Figure 2a (red axis label). The trends in the number of phonons generated and the maximum process temperature were similar because the phonons that were generated by colliding oxygen ions caused an increase in the material’s temperature. For Ion Beam 3, the PET substrate reached a temperature of 120 °C, which is much higher than the glass transition temperature of PET [26].

Plateau–Rayleigh instabilities, in which a one-dimensional fluid is broken up into droplets due to the minimization of interfacial surface tension, can occur in a liquid column [27,28,29]. When the solid PET fiber reached the glass transition temperature or higher, the fluidity of the fibers increased, and the fibers became agglomerated. The agglomerated fibers blocked the filter’s pores, causing a pressure loss and possible degradation of the filter performance. Optical microscope images of the filters are shown in Figure 2b–e. The agglomerated fibers were observed in the filter treated by Ion Beam 3 but not in the bare filter (without ion beam treatment) or the filters treated by Ion Beam 1 or 2.

### 3.2. Composition and Adhesion Properties of the Filters

The SEM and energy-dispersive X-ray spectroscopy (EDS) mapping images of the bare filter and the copper-coated filter treated with Ion Beam 2 are shown in Appendix A, respectively. The mass ratios of the major elements contained in each filter are shown in Appendix A. Carbon and oxygen comprise the vast majority of the bare filter, whereas the copper-coated filter contained copper and oxygen; the copper was evenly distributed on its surface (see Appendix A). Appendix A shows the optical microscope images of the surface of the copper-coated filters. Appendix A show that the appearance of the bare filter and the filters treated by Ion Beams 1 and 2 were similar. However, the filter fibers treated by Ion Beam 3 were agglomerated, as shown in Appendix A.

The strong adhesion between the filter fibers and the deposited copper film is essential because it can prevent copper nanoparticles (i.e., toxins) from entering the human body. The adhesion of the copper film deposited onto the filter fibers was evaluated using a T-peeling test with commercially available tape. This test confirmed that the peeling strengths of the filters treated with oxygen ion beams were higher than those of the bare filter, as shown in Figure 3. The bare filter’s peeling force per 10 mm of tape was 3.13 ± 0.05 N/cm, and the Ion Beam 2–treated filter exhibited the highest adhesion of 4.41 ± 0.11 N/cm.

Figure 4a–d show the results of the microscopic observations of the surfaces of the filters and the tape after it was peeled off, and Figure 4e shows a diagram of the evaluation of the tape. Figure 4a,b show that parts of the copper deposited onto the bare filter were released and transferred to the tape through cracks. This means that the adhesion between the copper film and the PET fibers was lower than the T-peeling result of 3.13 ± 0.05 N/cm would suggest. Figure 4c,d shows the filter treated by Ion Beam 1 and the tape peeled off it, indicating that copper did not transfer from the fiber to the tape. Appendix A show low-magnification images and SEM/EDS of the filter and tape surfaces after the T-peeling test, respectively. Among the elements analyzed through EDS in Appendix A, carbon is included in the PET filter and adhesive of tape, copper is in the coated copper film, and silicon is in the adhesive of tape. It was confirmed through EDS analysis in Appendix A that the tape’s adhesive was transferred to the copper thin film; no copper thin film was detached. The copper did not peel off the filters that underwent ion beam treatment; instead, the tape’s adhesive was transferred to the copper-coated filter surface. This indicates that the adhesion between the copper and the fibers in the filters that underwent ion beam treatment was significantly more robust than that in the bare filter. This is reflected in Figure 3, which shows that the filters treated with the ion beams exhibited a peeling force of 4.01 N/cm or more in the peeling test, compared to 3.13 N/cm for the bare filter.

### 3.3. Antibacterial and Antiviral Properties of the Copper-Coated Filter

The antibacterial properties of the copper-coated filter treated by Ion Beam 2 were investigated, according to the method described in Section 2.5, using the following bacteria: *Staphylococcus aureus* ATCC 6538, *Klebsiella pneumoniae* ATCC 4352, *Escherichia coli* ATCC 25922, and *Pseudomonas aeruginosa* ATCC 27853. The bare filter was used as the control sample. The antibacterial properties of the copper-coated filter are presented in Figure 5 and Appendix A. All of the bacteria grew to approximately 107 CFU/mL in the control sample. In contrast, the bacteria grew to 10 CFU/mL or less in the copper-coated filter. This implies a reduction in bacterial growth by more than 99.999% (a logarithmic value greater than 5), confirming that the copper-coated filter had antibacterial properties.

The SARS-CoV-2 evaluation system, composed of a copper-coated filter using a cylindrical device, is shown in Figure 1c. Figure 6a displays images of the bare filter and the copper-coated filter after spraying with SARS-CoV-2 aerosol, and Figure 6b shows images of the assay plates after being stained by crystal violet plaque. The copper-coated filter induced inactivation of the SARS-CoV-2 aerosol by more than 99.8% compared to the control group (as indicated in Figure 6b) as the detection limit (1.699 log PFU filter) was reached. By comparison, Hutasoit et al. reported that 96% of SARS-CoV-2 was inactivated on copper-coated stainless steel in two hours. In our previous work, we found that SARS-CoV-2 exposed to copper-coated masks for one hour was inactivated in a real-time polymerase chain reaction [12,19]. These results and those found in this study demonstrate that copper can eliminate viruses after short periods of physical contact.

## 4. Conclusions

In summary, an ion beam treatment technique was used to strongly adhere copper to the fibers found in a filter, the core component in masks and air purifiers. After treating the filter surface with an oxygen ion beam, strong adhesion of at least 4.01 N/cm was achieved between the copper and the filter fibers. Furthermore, we found that the ion beam–surface treatment process should be performed below the glass transition temperature of filter material to prevent damaging the filter fibers. The copper-coated filter developed in this study demonstrated a capability to remove bacteria and viruses remaining on its surface. However, there is a high possibility that bacteria and viruses not caught in the filter cannot be removed, and studies on the possibility of changing the antibacterial and antiviral properties by accumulated dust, adsorbed moisture, and oxidized copper on the surface when the filter is used for a long period have not been conducted. Nevertheless, the use of this copper coating filter can minimize the risk of secondary infections that may occur during the replacement and disposal of filters compared to conventional. Thus, copper-coated filters should be able to effectively and sustainably prevent infections caused by not only SARS-CoV-2 but also unknown viruses that may occur in the future. Moreover, unlike coating copper on the finished mask product that we previously reported, depositing copper on the filter fiber is advantageous because it provides a wider range of personal hygiene supplies utilization [19]. The copper coating filter developed in this study can be used for masks, filters for air purifiers, and building air conditioning filters, which can lead to a healthy life against airborne infectious diseases.

## Figures and Tables

**Figure 1 polymers-14-01007-f001:**
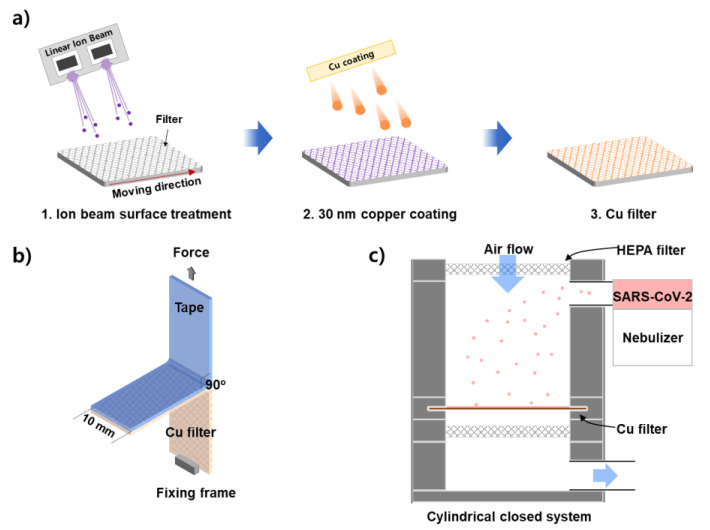
Diagrams of the (**a**) ion beam treatment and copper sputtering processes, (**b**) T-peeling test, and (**c**) closed system used to evaluate aerosol filters.

**Figure 2 polymers-14-01007-f002:**
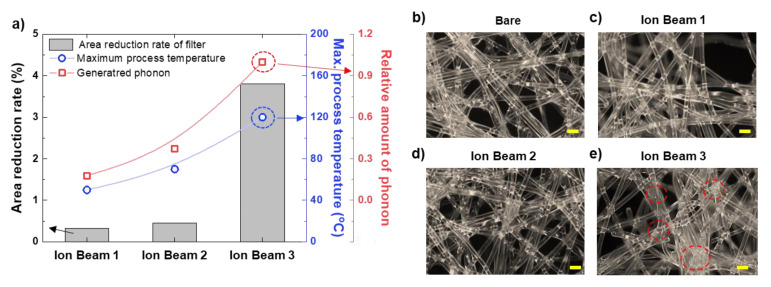
(**a**) Rate of change of area of the filter (black axis on the left), maximum process temperature reached during ion beam treatment (blue axis on the right), and the relative number of phonons generated as calculated by SRIM (red axis on the right) under ion beam irradiations of 1.45, 3.13, and 8.84 J/cm^2^. (**b**–**e**) Optical microscope images (scale bar: 100 µm) of the bare filter and the filters treated by Ion Beams 1, 2, and 3, respectively, under a dark field. The red-dashed circles show the areas where the fibers became agglomerated.

**Figure 3 polymers-14-01007-f003:**
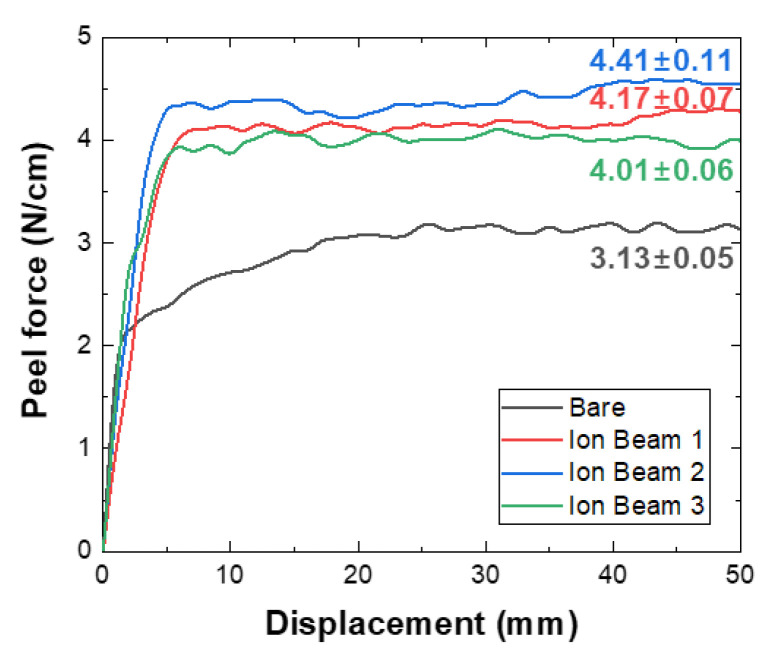
Peeling force per 10 mm of tape in the T-peeling test.

**Figure 4 polymers-14-01007-f004:**
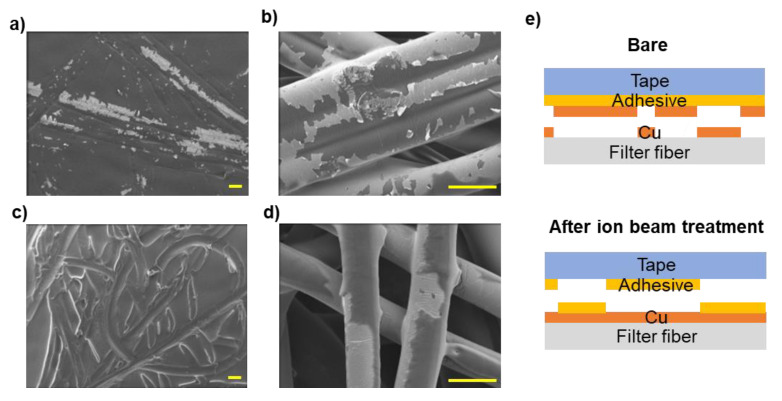
SEM images (scale bar, 50 micrometer) of the surface after peeling test (**a**) tape surface and (**b**) filter surface of bare, (**c**) tape surface and (**d**) filter surface treated under Ion Beam 1, and (**e**) schematic diagram of the peeling test result of the untreated (bare) and ion beam–treated specimens.

**Figure 5 polymers-14-01007-f005:**
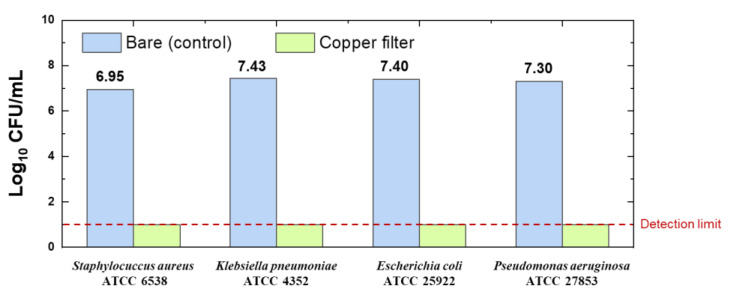
Bare filter’s (control) and copper-coated filter’s antibacterial properties against the following bacteria: *Staphylococcus aureus* ATCC 6538, *Klebsiela pneumoniae* ATCC 4352, *Escherichia coli* ATCC 25922, and *Pseudomonas aeruginosa* ATCC 27853.

**Figure 6 polymers-14-01007-f006:**
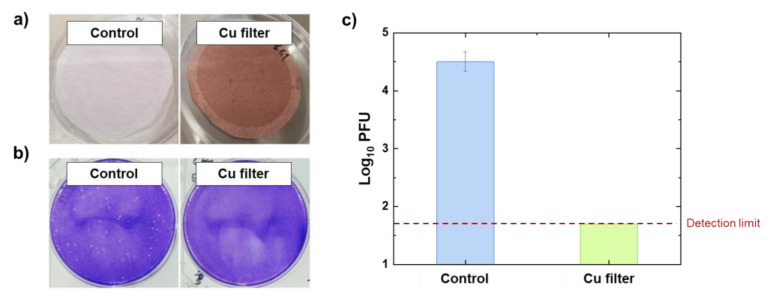
(**a**) Images of the bare filter and the copper-coated filter after spraying SARS-CoV-2 aerosol. (**b**) Images of the assay plates stained with crystal violet plaque. (**c**) Antiviral properties of copper-coated filter against SARS-CoV-2.

## Data Availability

No new data were created or analyzed in this study. Data sharing is not applicable to this article.

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
