# Peer review of "Sustainable Antibacterial and Antiviral High-Performance Copper-Coated Filter Produced via Ion Beam Treatment"

_polymers, 2022, doi:10.3390/polym14051007_

Round 1
Reviewer 1 Report
file attached

Author Response
Please see the attachment
In summary, detailed processes for sputtering and ion beam were added, and various English phrases were corrected. The English text was corrected with the help of professional English proofreading companies Editage and Grammarly.

Reviewer 2 Report
The authors have prepared filter surfaces coated with copper by an oxygen ion beam with a controlled process temperature. The filters coated with copper have been demonstrated that an inactivation of more than 99% of SARS-CoV-2 virus. Overall, this work can inspire more material design ideas for the filter. Therefore, I would like to recommend this work to publish in Polymers. Below are a few suggestions for the authors.
1. This paper would be more impressive if the author could check the filter (with and without coating with copper) before and after exposing the filter to bacteria by optical microscope or SEM. This can demonstrate the durability of the filter.
2. In the introduction “In contrast, copper is a sustainable antimicrobial material that has proven its ability to inactivate viruses”, more references could be cited to broaden the examples.
https://doi.org/10.3390/nano10061123
https://doi.org/10.3390/ijms20122924
Reviewer 3 Report
I have reviewed the manuscript by Jung et al., “Sustainable antibacterial and antiviral high-performance copper-coated filter produced via ion beam treatment” submitted to “Polymers” for publication. In this study, the authors have investigated the impact of an oxygen ion beam with a controlled process temperature to treat filter surfaces with copper. The manuscript has a little relevance to the scope of the polymers. Therefore, authors are suggested to justify its fitness in the scope of polymers.
The manuscript needs some major improvements; there are a few suggestions that authors may consider to improve it further:
The use of English language is reasonable, however, there are a number of punctuation and grammatical errors; that should be corrected and rephrased using academic English for a better flow of text for reader.
Abstract: is reasonable, however some of the key findings/results can be further added to improve the abstract.
Authors should make sure that all the abbreviations are defined at their first appearance in the text and use abbreviations afterword. Some of the abbreviations used once in the abstract should be removed. For example: SARS-CoV-2 should be defined.
Introduction is very comprehensive and detailing all the background information and rationale of the study.
Line 34-42: all the information here needs updating with the current statistics.
Line 72: replace “discovered” with “investigated”
In academic writing, words such as "they” are not used frequently; try to replace it with appropriate words.
Please add manufacturer details (City country) for the materials and equipment mentioned in the materials section.
Results and discussion sections are comprehensive, coving all the results and their discussion.
What are the limitations of the study; limitation and further work to overcome such limitations should be included in the last part of the discussion section.
Round 2
Reviewer 1 Report
see attached file

Reviewer 3 Report
I carefully checked the manuscript. The authors responded to all the comments and the manuscript may be accepted for publication.